# Iodine Intake and Related Cognitive Function Impairments in Elementary Schoolchildren

**DOI:** 10.3390/biology11101507

**Published:** 2022-10-14

**Authors:** Helga B. Bailote, Diana Linhares, Célia Carvalho, Susana Prazeres, Armindo S. Rodrigues, Patrícia Garcia

**Affiliations:** 1Faculty of Sciences and Technology, University of the Azores, 9501-801 Ponta Delgada, Portugal; 2IVAR, Institute of Volcanology and Risks Assessment, University of the Azores, 9501-801 Ponta Delgada, Portugal; 3Faculty of Social and Human Sciences, University of Azores, 9500-321 Ponta Delgada, Portugal; 4CINEICC, Cognitive and Behavioral Centre for Research and Intervention, Faculty of Psychology and Educational Sciences, University of Coimbra, 3000-115 Coimbra, Portugal; 5Laboratory of Endocrinology, Department of Clinical Pathology, Portuguese Institute of Oncology of Lisbon Francisco Gentil, E.P.E., 1099-023 Lisbon, Portugal; 6cE3c, Centre for Ecology, Evolution and Environmental Changes, Azorean Biodiversity Group, University of the Azores, 9501-801 Ponta Delgada, Portugal

**Keywords:** iodine deficiency, intelligence, WISC-III, CPM, cognitive function

## Abstract

**Simple Summary:**

This is the first survey to show throughout the application of Wechsler Intelligence Scale for Children complete form, how the adverse effects of moderate iodine-deficiency negatively impair cognitive function in schoolchildren. The results obtained suggest differences in the cognitive profile of the children with moderate iodine-deficiency, denouncing several changes in specific mental functions which could attaining their full intellectual potential, resulting in a differentiated profile of intellectual development and may have some clinical value. Working memory, in our study seriously compromised in moderate iodine-deficient schoolchildren, is used to process, and store information during complex and demanding activities and it supports many activities that iodine-deficient children usually engage in at school. Given the heavy working memory demands of classroom instructions and daily activities, it is perhaps unsurprising that one of the key characteristics of the schoolchildren with moderate iodine-deficiency is poor educational attainment, thereby; iodine deficiency represents a significant risk factor for poor educational progress. For policy makers and governments, it is imperative to act against this iodine-deficiency epidemiology in the most urgent terms. Cognitive impairments represent huge costs for the region, affecting local productivity, economy, and the region´s development and economic potential.

**Abstract:**

Iodine deficiency, the most common cause of preventable mental impairment worldwide, has been linked to poorer intellectual function in several studies. However, to our knowledge, no studies have been performed in moderate iodine-deficient schoolchildren using the complete form of Wechsler Intelligence Scale for Children (WISC-III; Portuguese version). The main purpose of this study was to ascertain whether moderate iodine deficiency would affect the cognitive function of schoolchildren (7–11 years old; 3rd and 4th grades). Raven’s Colored Progressive Matrices (CPM; Portuguese version) were used for measuring the intelligence quotient (IQ) of the total population (n = 256; median UIC = 66.2 μg/L), and the WISC-III was used to study two selected subgroups: one moderately iodine-deficient (n = 30) and the other with adequate iodine intake (n = 30). WISC-III was shown to be the prime instrument for cognitive function assessment among moderate iodine-deficient schoolchildren; this subgroup had a Full-Scale IQ 15.13 points lower than the adequate iodine intake subgroup, with a magnitude effect of d = 0.7 (*p* = 0.013). Significant differences were also registered in 6 of the 13 Verbal-Performance IQ subtests. Moderate iodine deficiency has a substantial impact on mental development and cognitive functioning of schoolchildren, with significant impairment in both Performance IQ and Verbal IQ spectrum, adversely impacting their educational performance.

## 1. Introduction

Iodine deficiency (ID) is one of the most serious public health issues worldwide [1,2], recognized as the most common cause of preventable brain damage in the world and generally agreed as the primary cause of iodine deficiency disorders (IDD) [3,4], even in mild or moderate iodine deficiency status [5]. Low concentrations of thyroxine can adversely affect fetal brain development and, subsequently, child and adult cognitive function [6]. During pregnancy, severe iodine deficiency can cause cretinism, whereas mild-to-moderate ID impairs the neurocognitive function of the offspring [7,8]. Andersson et al. (2010) [9] estimated in 2011 that 241 million school-aged children were at risk of IDD, which extrapolates to 1.88 billion people globally. According to Zimmermann (2010) [10], half of the population from continental Europe are mildly iodine-deficient.

People living in areas affected by severe iodine deficiency may have an intelligence quotient (IQ) of up to 13.5 points below that of those from comparable communities in areas where there is no iodine deficiency [11]. In a meta-analysis using Medline (1980–November 2011), carried out by Bougma et al. (2013) [12], iodine-deficient children scored 6.9 to 10.2 IQ points lower than iodine-replete children. The most convicting evidence comes from a randomized controlled trial conducted by Zimmermann et al. (2006) [13] in moderate iodine-deficient (urinary iodine concentration = 43 μg/L) Albanian children (10–12 years old). In this study, the authors found that children in the iodine-replete group performed significantly better on tests of cognitive function than the children in the placebo group. On the other hand, Gordon et al. (2009) [14] observed that iodine supplementation improved perceptual reasoning in mildly iodine-deficient children (10–13 years old) from New Zealand.

The Wechsler Intelligence Scale for Children (WISC) is considered the most powerful and sensitive test for intelligence measure. It is regularly used in many areas of psychology and child development as a general measure of intelligence. Nevertheless, for studies regarding iodine deficiency and cognitive function it has only been applied in its abbreviated form [15] or by only using a small number of subtests [13,14,16]. The WISC-III (Portuguese version) [17] is an individually administered test of intellectual ability for children aged 6 to 16 years. It comprises ten mandatory and three optional subtests that combine to yield Verbal, Performance, and Full-Scale IQ. Raven’s Colored Progressive Matrices (CPM) are frequently used as a complementary nonverbal IQ measure [13,14]. The CPM, already standardized for the Portuguese population between 6 and 11 years old [18,19], is one of the most used instruments to assess non-verbal intelligence and has been used extensively as a “culture-fair” test of intelligence. Both these measures of cognitive functioning provide useful information about children’s intellectual abilities and cognitive strengths and weaknesses.

Intelligence in childhood, assessed by psychometric cognitive tests, is a strong predictor of several important life outcomes, including educational attainment, income, health, and lifespan [20,21]. Although iodine-deficient intake has been linked to poorer intellectual function in several studies, none was developed in Portugal´s worst-case scenario, registered in the Azores archipelago, where 78.4% of the schoolchildren had a urinary iodine concentration (UIC) <100 μg/L [22]. The association between ID and cognitive impairments, using the CPM and the WISC as instruments for intelligence measure, have been reported in New Zealand [14] and Albanian surveys [13]. However, not all the IQ subtests for WISC have been used in these studies, leaving some questions to answer, such as: which other cognitive functions could be compromised in iodine-deficient schoolchildren and what is the magnitude of the effect of iodine deficiency in IQ?

To our knowledge, this is the first study that examines the effects of iodine-deficient intake using both CPM and the complete form of the WISC-III in schoolchildren. In this work, the cognitive function of schoolchildren (7–11 years old) from Terceira Island (Azores) and its association with iodine deficiency will be assessed in an observational study (n = 256; only CPM was used) and in a case-control study, using two subgroups selected from the previous: one with moderate iodine intake (n = 30) and the other with adequate iodine intake (n = 30). Both CPM and WISC-III were used.

By using this approach, the following objectives were considered: (i) to analyze iodine-deficient schoolchildren performance on CPM; (ii) to analyze moderate iodine-deficient vs. adequate iodine intake children scores and effect size using WISC-III Full-Scale IQ and Verbal-Performance IQ subtests; (iii) to assess which IQ measure provides more useful information about the presence of specific cognitive disabilities in iodine-deficient schoolchildren; (iv) to identify specific cognitive disabilities and related neuropsychological impairments linked to moderate iodine deficiency in schoolchildren.

## 2. Materials and Methods

### 2.1. Study Area

This study was carried in Terceira Island, one of the largest islands of the Azores Archipelago, located in the middle of the North Atlantic Ocean (Figure 1A).

### 2.2. Study Population and Assessment of Iodine Intake

Sixteen elementary schools from Terceira Island (Figure 1B) were selected. Clarification meetings were carried with each school coordinator, and children from the 3rd and 4th grades were recruited after a brief presentation of the project in their classrooms. A randomized survey was carried in 256 schoolchildren (aged between 7 and 11 years old), representing 23% of the 3rd and 4th scholar grade population. Children´s legal guardians were asked to answer a sociodemographic and dietary habits questionnaire. Only children that regularly had lunch at the school canteen and had no known recent history of psychological evaluation, intellectual disability, or confirmed thyroid conditions were considered for the study. Iodine intake data was recorded using a modified form from validated iodine-specific food frequency questionnaires [23,24,25]. The Regional Directorate of Education allowed this study, and the parents or legal guardians of all participants provided informed written consent in compliance with the Helsinki Declaration and Oviedo Convention. Ethical approval for this study was obtained from the Ethics Boards of Santo Espírito da Ilha Terceira Hospital, EPER, Angra do Heroísmo.

### 2.3. Urinary Iodine Concentration Assessment

Iodine status was assessed by measuring urinary iodine concentration. From each participant, one spot-urine sample (first urine in the morning) was collected, capped in sterile plastic tubes containers, encoded, and preserved in freezing conditions until use. Urinary iodine (UI) was measured by a fast colorimetric method [26] in the Laboratory of Endocrinology of Portuguese Institute of Oncology in Lisbon. Iodine status in schoolchildren was classified in the following WHO´s ranks, according with the UIC [27]: severe iodine deficiency, 0–19 μg/L; moderate iodine deficiency, 20–49 μg/L; mild iodine deficiency, 50–99 μg/L; adequate iodine intake, 100–199 μg/L; more than adequate iodine intake, 200–299 μg/L; excessive iodine intake, ≥300 μg/L.

### 2.4. Cognitive Function Assessment

#### 2.4.1. Raven´s Colored Progressive Matrices (CPM)—Sets A, AB, B

The CPM, standardized for the Portuguese population between 6 and 11 years old, comprises 36 items, in which the subject is required to indicate the correct target among six alternatives. Grouped into three sets (A, AB, B) of 12 items in each one, the CPM solving requires different abilities, namely: A Series, requiring visuo-perceptual ability; AB Series, symmetric ability; and B Series, conceptual and analogic thought abilities [18,19,28]. The CPM was applied to all schoolchildren (n = 256) separated in small groups, never exceeding eight participants for session. Each session took approximately 20 to 45 min and was carried out by one-trained psychologist without knowledge of each child’s individual iodine intake status. Each child received one answer sheet and the illustrated test booklet. The first three items were used for practice and the interviewer proceeded with the next items whenever the object of the task was clear [19]. Raven´s raw scores obtained from tests were subsequently scaled for age. Results are interpreted with a percentile scale between 1–99 [29]: Percentile scores ≥95, Level I, very superior intelligence; ≥75, Level II, superior intelligence (90–94 Level II+/-); 25–75, Level III, normal or average intelligence (50–75 Level III+; 25–50 Level III-); ≤25 Level IV, below average (≤10 Level IV-); ≤5 Level V, feeble mindedness/very low.

#### 2.4.2. Wechsler Intelligence Scale for Children—3rd Edition (WISC-III)

Full-Scale IQ, with 10 core and 3 additional subtests, from the Wechsler Intelligence Scale for Children, third edition (WISC-III), Portuguese standardized edition [30], normed on a representative sample of 1354 children, was used to assess cognition in 2 subgroups of schoolchildren. These subgroups, with 30 children each, were selected from the study population (n = 256): a subgroup with adequate iodine intake (UIC 100–199 μg/L) and a subgroup with moderate iodine deficiency (UIC 20–49 μg/L). The WISC-III [31] is an individually administered intelligence test, including 13 subtests, for children between the ages of 6 and 16 years, that measures different intellectual abilities, having outputs of several scores. Verbal IQ and Performance IQ scores are assessed separately, along with their sum (Total IQ score), which is an index of general intellectual functioning. All subtests (briefly described on Appendix A), were chosen based on 3 criteria: (i) the aspect of cognition assessed test by test; (ii) the likelihood of the tests to respond to neuropsycho-intellectual deficits in mild-moderate ID children based on previous research [13,14,32,33,34]; (iii) the inexistence of studies comparing intellectual performance of children with moderate iodine deficiency and children with adequate iodine intake with the WISC-III Full-Scale IQ. The testing was conducted at school, in one session of approximately 60 to 120 min, by four trained psychologists blind to children’s iodine status. All measures were administered in a standard order. Qualitative comments were made as a function of the score scaled for age [17]; IQ scores ≤69 reveal a feeble-mindedness; 70–79 borderline deficiency in intelligence; 80–89 low average intelligence; 90–109 normal or average intelligence; 110–119 high average intelligence; 120–129 superior intelligence; over 130 reveals very superior intelligence. 

### 2.5. Statistical Analyses

For the entire population (n = 256), differences in CPM total and percentile scores between UIC rank groups were analyzed by Kruskal–Wallis. Chi-Square test was used to compare IQ level across UIC ranks and location area. For the two subgroups (n = 30, each), *t*-test was used to compare the following continuous variables: age; CPM total and percentile scores; WISC-III Total IQ, Verbal IQ, Performance IQ scores, and outcomes of each WISC-III cognitive subtest. Chi-Square test was used to compare the following categorical variables: location area (urban vs. rural); gender (male vs. female); scholar grade (3rd vs. 4th); failure to pass year (yes vs. no); reading habits (yes vs. no); environmental tobacco smoke exposure (yes vs. no); employed parents (yes vs. no); parents with learning difficulties at school (yes vs. no); parents expectations for children accomplishing university level studies (yes vs. no); iodine supplements in pregnancy (yes vs. no). All statistical analyses were performed using SPSS for windows [35], and the level of statistical significance was set at *p* < 0.05.

## 3. Results

### 3.1. Characteristics of the Study Population and Study Subgroups

Children’s mean age was 9.04 ± 1.019 y, and 70.8% were 8 or 9 y old; gender was evenly represented (48.8% were boys). The study population was either from rural (45.7%) and urban (54.3%) areas, recruited from the third (n = 135) and fourth (n = 121) scholar degree. 

The two subgroups are socially, economically, and educationally similar (Table 1).

### 3.2. Urinary Iodine Concentration (UIC)

The median UIC was 66.2 μg/L, corresponding to a mild iodine deficiency (UIC 50–99 μg/L). The interquartile range was 42.51 (37.38–79.89 as 25th–75th percentile). Additionally, 78.9% of the children had an insufficient iodine intake (UIC <100 μg/L), with 35.5% having a UIC <50 μg/L. Adequate iodine intake (UIC 100–199 μg/L) was present in 19.9% of the population (Figure 2). 

### 3.3. Cognitive Function Assessment

#### 3.3.1. Raven´s Colored Progressives Matrices (CPM)—A, AB, B

No significant differences in IQ level by location area (rural vs. urban) were observed (χ2(4) = 9.036, *p* = 0.06) and, despite randomization, also for iodine status groups (χ2(16) = 15.239, *p* = 0.507). CPM IQ total and percentile scores had no significant differences between iodine status groups (*p* = 0.117 and *p* = 0.384, respectively) (Appendix A). The study subgroups showed no significant differences in CPM IQ total (t(58) = 1.961, *p* = 0.055) and percentile scores (t(58) = 1.758, *p* = 0.089) between moderate iodine deficiency and adequate iodine intake (Appendix A). However, a clear tendency for schoolchildren to score lower with moderate iodine deficiency can be observed (Figure 3). 

#### 3.3.2. Wechsler Intelligence Scale for Children—3rd Edition (WISC-III)

Full-Scale IQ differed significantly between the subgroups with moderate iodine deficiency and adequate iodine intake (mean 86.67 vs. 101.80, t(58) = 2.568, *p* = 0.013). Schoolchildren with moderate iodine deficiency had 15% lower scores, and an inferior difference of 15.13 IQ points, when compared with the adequate iodine intake group (Figure 4).

Moreover, the mean Full-Scale IQ of moderate iodine deficiency subgroup is below the cutoff point of 90 specified by Wechsler (2003). Seventy-five percent of all cases with Total IQ below average (<90) were scored by schoolchildren with moderate iodine deficiency. Cohen´s d shows a very large effect size of iodine intake status on Full-Scale IQ (d = 0.7) (Table 2). The moderate iodine deficiency subgroup exhibited significantly lower scores in the Full-Scale, Verbal-, and Performance IQ (86.67, 44.50, and 42.17, respectively), compared to those from the adequate iodine intake subgroup (101.80, 52.50, and 49.30, respectively) (Table 2). 

The WISC-III age-standardized IQ percentiles scores followed the same trends: significant differences in Full-Scale IQ (mean 29.66 vs. 50.73, t(58) = 2.651, *p* = 0.010), and also on Verbal IQ (mean 36.29 vs. 58.53, t(58) = 2.786, *p* = 0.007), with larger effect sizes in Information (*p* = 0.024, d = 0.6), Similarities (*p* = 0.014, d = 0.7) and Digit Span (*p* = 0.006, d = 0.7) (Table 3). Similar results were obtained with Performance IQ for Picture Completion (*p* = 0.007, d = 0.7), Picture Arrangement (*p* = 0.027, d = 0.6), and Block Design (*p* = 0.026, d = 0.6) (Table 3).

## 4. Discussion

Several studies in populations with ID have shown the association between iodine deficiency status and poor mental and psychomotor development [12,38,39]. However, while progress has been made in the understanding of this topic, there are still many gaps examining the role of iodine and cognition [40]. Our results are comparable with the most convicting evidence from two randomized controlled trials conducted by Zimmermann et al. (2006) [13] and Gordon et al. (2009) [14]; both these surveys used the CPM and some subtests of WISC for cognitive function assessment. The present study revealed that iodine status did not significantly affect the CPM IQ total and percentile scores, either for the whole studied group or when comparing adequate iodine intake and moderately iodine-deficient subgroups. However, analyzing the distribution of IQ levels in both subgroups, a tendency is seen for children from the moderate iodine-deficient subgroup to score lower, in accordance with Zimmermann et al. (2006) [13], and picture concepts and matrix-reasoning subtest on perceptual reasoning assessment in Gordon et al.’s (2009) [14] study. It is likely that these effects will be even more pronounced in individuals with severe ID, compared to those with moderate-to-mild ID, especially as the effects of less severe ID are nuclear [40]. Despite CPM having been extensively used as a “culture-fair” intelligence measure, our results reveal its poor sensitivity as a tool to analyze the cognitive effects for the whole ID population spectrum. The main sensitivity and discriminatory results obtained by the complete form of WISC-III make this measure the key instrument to identify specific cognitive impairments in ID schoolchildren. In our study, the moderate iodine-deficient schoolchildren subgroup scored a mean of 86.67 points in Full-Scale IQ, fitting in the 80–89 interval of IQ scores that reveals low average. In contrast, adequate iodine intake schoolchildren scored a mean of 101.80 points in Full-Scale IQ, which is in the normal or average intelligence IQ range (90–109) [17,31]. Moderate iodine-deficient schoolchildren scored 15.13 points lower than the adequate iodine intake subgroup in Full-Scale IQ, representing an IQ reduction of 15%. Although not using the WISC, the results obtained by Bougma et al., (2013) [12] also translate into IQ points lower in ID children compared with iodine replete children (6.9 to 10.2 vs. 15.13 in the present study). These lower IQ results have broad economic and social cost implications because intelligence affects wellbeing, income, and education outcomes [41]. 

In our study, schoolchildren with moderate iodine deficiency revealed significant deficits in 6 of the 13 WISC-III subtests, on both Verbal and Performance IQ, when compared with the adequate iodine intake subgroup: larger effect sizes are observed in Information, Similarities, Digit Span, Picture Completion, Picture Arrangement, and Block Design subtests profile scores.

Low scores in Information and Similarities general ability show ID’s predisposition to impair cognitive functioning at the verbal comprehension dimension. The application of this knowledge involves verbal concept formation, reasoning, and expression as a measure of crystalized intelligence [17,42]. 

Digit Span subtest is designed as a measure of auditory short-term memory, sequencing skills, attention, and concentration that requires storage and retrieval of information through immediate auditory recall. Forward designs involve tasks such as attention, auditory processing, encoding, rote learning, and memory. The backward designs involve mental manipulation, visuospatial imaging, transformation of information, and working memory. Shifting from forward to backward requires cognitive flexibility and mental alertness [17,42]. Working memory is an essential component of fluid reasoning and other higher order cognitive processes. It measures the child´s ability to register, maintain, and manipulate visual and auditory information in conscious awareness. Both verbal comprehension and working memory subtests are the best WISC-III/WISC-IV predictors of reading ability [17,42]. Therefore, according to our results, moderate ID may well compromise schoolchildren reading and cognitive processes. Additionally, the low scores registered in Digit Span subtest show a predisposition to cognitive function impairment at the auditory and working memory dimensions in the moderate ID subgroup. 

The Picture Completion subtest was designed to measure visual perception and organization, concentration, and visual recognition of essential objects details. Incapability to attend and concentrate, poor visual memory, inability to note detail, poor reality testing, anxiety, or even depression, are possible causes of significant low scores [17,42]. According to Lezak (1995) [43], Picture Completion consistently demonstrates resilience to the effects of brain damage. It would be useful to conduct future research to confirm these findings and compare this with the role of the trait emotional intelligence, as well as to ascertain the reversibility or permanence of the changes seen. 

Low scores in Picture Arrangement usually result from an inability to sequence, poor social knowledge, inadequacy to note action and plan of action’ lack of social skills, impulsiveness, inability to note detail, incapacity to respond to time pressure, poor reality testing, poor visual–motor coordination, and anxiety and depression [42]. 

Low scores in nonverbal Block Design subtest show ID predisposition to general ability impairment at the visual spatial dimension [17]. Therefore, a child´s ability to evaluate visual details and to understand visual spatial relationships to construct geometric designs from a model could be compromised in ID schoolchildren. 

Cognitive and Math-related processes involve attention [44,45], visual–spatial skills [46], and working memory [47]. According to the observed low scores of both Digit Span and Block Design subtests, moderately ID schoolchildren have a larger predisposition to have difficulties in mathematics.

Symbol Search is the unique subtest in common with Gordon et al.’s (2009) [14] and Zimmermann et al.’s (2006) [13] studies. In contrast with the findings in our study and Gordon et al.’s (2009) [14] studies, Zimmermann et al. (2006) [13] registered a significant improvement for Symbol Search in Albanian children with moderate ID and supplemented with iodine compared with the children in the placebo group. One possible explanation for our results, proposed by Gordon et al., (2009) [14], is that symbol search has a testing format more familiar than the other subtests, in this case, to New Zealand and Portuguese children. Nevertheless, both Gordon et al. (2006) [14] and Zimmermann et al. (2006) [13] also suggested that ID children have memory impairment. Therefore, some of these results need to be carefully interpretated and compared. For example, in the study carried in Albania, the WISC and the CPM were not translated and standardized, which could somehow have biased the Albanian schoolchildren scores. 

A limitation of our study is the fact that subtle effects observed on cognitive abilities may have been mediated by other not controlled variables, e.g., genetics, birth weight, exposure to environmental contaminants, infections, prenatal stress, and early life stress [48]. Additionally, intervention studies usually provide a more powerful indicator of the relation between iodine and cognition [40] than the observational ones.

Nevertheless, many strengths remain present in our observational study: first, the much representative sample size and exceptional compliance of the schoolchildren over the study period; second, it is the first study in Portugal, namely in Azores archipelago, and in Terceira island (considered a mild ID region, [49]), which compares iodine status with cognitive functioning in a large and representative population of schoolchildren (23% of the 3rd and 4th grades schoolchildren from Terceira Island were studied); third, a clear highlight, because this is the first, to our knowledge, measuring cognitive function in ID schoolchildren with the complete form of the WISC-III, contributing with several results that allows a wide range of cognitive functions to be examined, with sensitive scoring to specific brain structures impairments.

Our work clearly reveals that ID has a substantial impact on mental development and cognitive functioning, with significant impairment in both Performance IQ and Verbal IQ spectrum, which lead us to propose possible impairment predominantly on the hippocampus, prefrontal cortex, caudate nucleus, and auditory pathways, affecting memory, attention, and language functions [50,51]. However, the human brain is a bizarre device [52] and it is still unclear how communication between brain networks responds to changing environmental demands during complex cognitive processes [53] as working memory tasks. For that reason, to prevent and treat ID, we must clearly understand how and when iodine affects the brain and cognitive development and whether these effects on cognitive functions persist over the longer term and have an adverse impact on educational performance. 

## 5. Conclusions

The results of this study reveal differences in the cognitive profile of the children with moderate iodine deficiency. Several changes in specific mental functions which could limit attaining full intellectual potential were observed, resulting in a differentiated profile of intellectual development which may have some clinical value. Working memory, seriously compromised in moderate iodine-deficient schoolchildren, is used to process and store information during complex and demanding activities, supporting many actions that children usually engage in at school. As iodine deficiency represents a significant risk factor for poor educational progress, the action of policy makers and governments is imperative against this iodine-deficiency epidemiology in the most urgent terms. Implementing partnership policies, iodine public health programs and school intervention projects in iodine-deficient areas should be implemented, along with specific adaptations according to each scenario.

## Figures and Tables

**Figure 1 biology-11-01507-f001:**
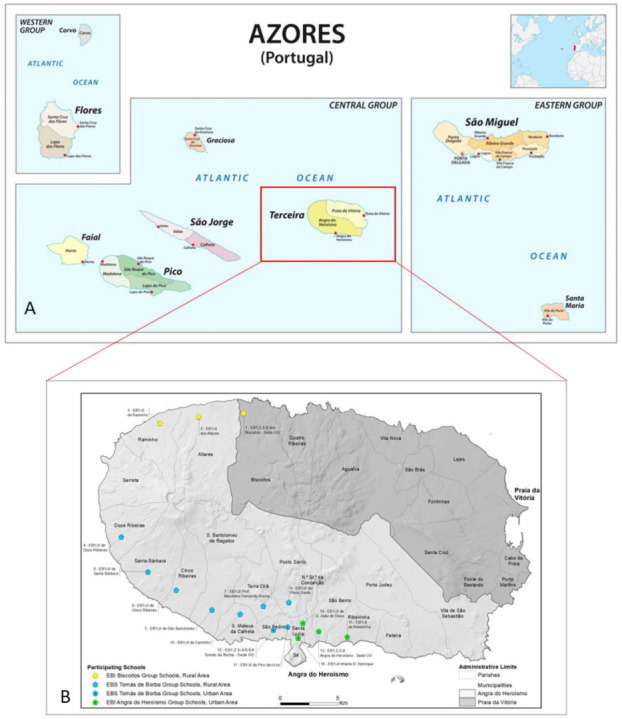
Location maps of the Azores archipelago (**A**) and Terceira Island (**B**). The participating schools are represented by dots (green, blue, and yellow).

**Figure 2 biology-11-01507-f002:**
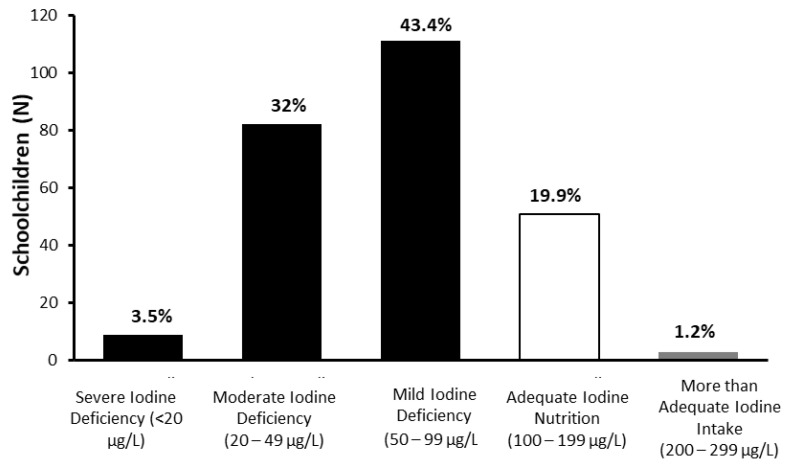
Schoolchildren distribution according to iodine status (median UIC= 66.2 μg/L, n = 256).

**Figure 3 biology-11-01507-f003:**
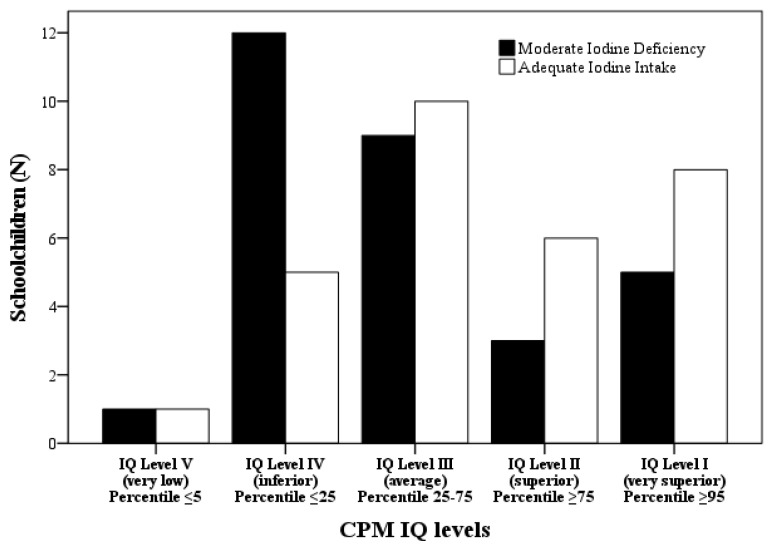
Raven´s Colored Progressive Matrices (CPM) IQ for Moderate Iodine Deficiency (UIC 20–49 μg/L, n = 30) and Adequate Iodine Intake (UIC 100–199 μg/L, n = 30) subgroups.

**Figure 4 biology-11-01507-f004:**
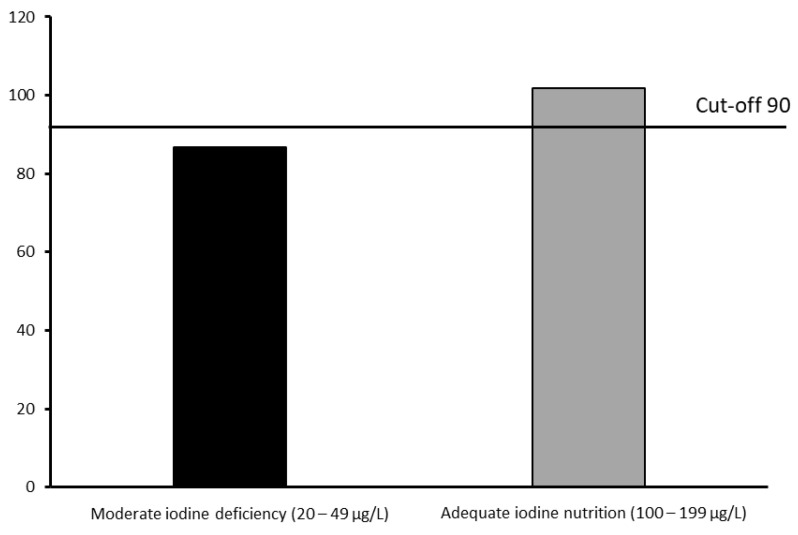
Wechsler Intelligence Scale for Children—Third Edition (WISC-III) Full-Scale IQ (mean) according to iodine status: Moderate Iodine Deficiency (UIC 20–49 μg/L, n = 30) vs. Adequate Iodine Intake (UIC 100199 μg/L, n = 30) subgroups.

**Table 1 biology-11-01507-t001:** Characteristics of the two studied subgroups: Moderate Iodine Deficiency vs. Adequate Iodine Intake.

Characteristic	Moderate Iodine-Deficient (UIC 20–49 μg/L, n = 30)	Adequate Iodine Intake (UIC 100–199 μg/L, n = 30)	*p*-Value ^a^
*General characteristics*			
Age, y	9.03 ± 1.066	8.90 ± 1.029	0.717
Gender, male	15 (50.0%)	16 (53.3%)	0.796
Scholar Grade, 3rd	22 (73.3%)	14 (46.7%)	0.035
*Study characteristics*			
Failure to pass year, no	20 (66.7%)	24 (80%)	0.561
Reading habits, no	15 (50%)	12 (40%)	0.549
Environmental tobacco smoke exposure, no	22 (73.3%)	17 (56.7%)	0.139
*Parents*			
Employed	21 (70%)	20 (66.7%)	0.500
Difficulties at school, no	8 (26.7%)	14 (46.7%)	0.390
Scholar grade			0.204
≤4	13 (43.3%)	8 (26.7%)	
5–9	8 (26.7%)	12 (40%)	
9–12	6 (20%)	3 (10%)	
>12	3 (10%)	7 (23.3%)	
Expectations for their children, scholar grade >12	20 (66.7%)	21 (70%)	0.540
Iodine supplements in pregnancy, no	28 (93.3%)	26 (86.7%)	0.335

All data are reported as the number of subjects (%) except for age (mean ± SD); **^a^** *t*-test (age); *χ*2 test (categorical variables).

**Table 2 biology-11-01507-t002:** WISC-III IQ full scores and age-standardized IQ percentiles scores in schoolchildren for moderate iodine deficiency and adequate iodine intake subgroups.

	Iodine Status				
Measure	Moderate Iodine Deficiency (UIC 20–49 μg/L, n = 30) *Score* ^1^	Adequate Iodine Intake (UIC 100–199 μg/L, n = 30) *Score* ^1^	*p* ^2^	Cohen´s *d* ^5^	Cohen´s Standard ^6^
**IQ full scores**					
*Full Scale IQ*	86.67 ± 23.042	101.80 ± 22.605	0.013	0.7 (43%)	Very Large
*Verbal IQ* ^3^	44.50 ± 13.612	52.50 ± 13.216	0.024	0.6 (38.2%)	Large
*Performance IQ* ^4^	42.17 ± 13.447	49.30 ± 11.532	0.031	0.6 (33%)	Large
**Age-standardized IQ percentiles scores**					
*Full Scale IQ*	29.66 ± 29.704	50.73 ± 31.796	0.010	0.7 (43%)	Very Large
*Verbal IQ* ^3^	36.29 ± 30.913	58.53 ± 30.937	0.007	0.7 (43%)	Very Large
*Performance IQ* ^4^	32.08 ± 32.104	47.07 ± 31.341	0.072	0.5 (33%)	Large

^1^ Mean ± SD; ^2^ Between-group differences: IQ full scores (*t*-test); Age-standardized IQ percentiles scores (1way ANOVA); ^3^ Based on five subtests: information, similarities, arithmetic, vocabulary and comprehension; ^4^ Based on five subtests: picture completion, picture arrangement, block design, object assembly and coding; ^5^ Measure of the magnitude of iodine intake status effect (mean difference between the two groups, divided by the pooled standard deviation) [36]; ^6^ Cohen´s d: 0.5–0.7—large effect sizes; 0.7–0.9—very large effect sizes [37].

**Table 3 biology-11-01507-t003:** Final mean percentile scores for each individual WISC-III cognitive subtests of verbal and performance IQ in schoolchildren for moderate iodine deficiency and adequate iodine intake subgroups.

Measure	Iodine Status				
	Moderate Iodine Deficiency (UIC 20–49 μg/L, n = 30) Score ^1^	Adequate Iodine Intake (UIC 100–199 μg/L, n = 30) Score ^1^	*p* ^2^	Cohen´s *d* ^3^	Cohen´s Standard ^4^
** *Verbal IQ* **					
*Information*	7.53 ± 2.80	9.23 ± 2.86	0.024	0.6	Large
*Similarities*	9.87 ± 3.33	12.20 ± 3.82	0.014	0.7	Very Large
*Arithmetic*	10.00 ± 2.96	10.73 ± 2.63	0.314		
*Vocabulary*	8.80 ± 3.70	10.27 ± 3.90	0.141		
*Comprehension*	8.30 ± 4.16	10.07 ± 3.27	0.073		
*Digit Span*	8.20 ± 3.18	10.40 ± 2.84	0.006	0.7	Very Large
** *Performance IQ* **					
*Picture Completion*	8.57 ± 3.68	11.27 ± 3.81	0.007	0.7	Very Large
*Coding*	7.57 ± 3.42	7.63 ± 2.80	0.934		
*Picture Arrangement*	9.13 ± 3.42	11.13 ± 3.40	0.027	0.6	Large
*Block Design*	8.40 ± 3.66	10.47 ± 3.35	0.026	0.6	Large
*Object Assembly*	8.50 ± 3.71	8.80 ± 3.27	0.741		
*Symbol Search*	7.33 ± 3.28	8.20 ± 3.33	0.314		
*Mazes*	8.40 ± 3.23	9.93 ± 4.06	0.111		

^1^ Mean ± SD; ^2^ Between-group differences (*t*-test); ^3^ Each sub-test in both Verbal and Performance Scales has a range from 1 to 19, with scores between 8 and 12 considered average. Measure of the magnitude of iodine intake status effect (mean difference between the two groups dividing by the pooled standard deviation); ^4^ Cohen´s d: 0.5–0.7—large effect sizes; 0.7–0.9—very large effect sizes [37].

## Data Availability

The datasets generated and/or analyzed during the current study are available in the University of the Azores repository, [https://repositorio.uac.pt/handle/10400.3/3842].

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
