# Peer review of "Iodine Intake and Related Cognitive Function Impairments in Elementary Schoolchildren"

_biology, 2022, doi:10.3390/biology11101507_

Round 1
Reviewer 1 Report
Comments to the Author:
1. The number of children recruited for this study was 256 while only 60 students were chosen for further study. On what criteria is justified that 30 students in each sub-set is adequate for the evaluation of data and not the total 256 numbers required?
2. In the sub-set, especially the moderate iodine deficiency, are there any data demonstrating any gender differences in performance? Any data presented will be warranted.
3. Fig 2, why moderate iodine deficiency (20-49 ug/L) was chosen and not the mild iodine deficiency (50-99 ug/L) group while your Median UIC was 66.2 ug/L? You have already 43.4% (n=111) of student who falls into the mild iodine deficiency category (Fig 2), in essence, this is the group that needs more attention to show any impairment due to insufficient iodine nutrition.
4. Would it be appropriate to assign a UIC value to each CPM IQ level in Fig 3 or data overlapping each other as the data set is small, n=30?
5. Tables 2 and 3 should have gender data included if it does not cause too much extra analysis. Parents, educators, and policymakers are all interested to ask this question.
Author Response
- The number of children recruited for this study was 256 while only 60 students were chosen for further study. On what criteria is justified that 30 students in each sub-set is adequate for the evaluation of data and not the total 256 numbers required?
The CPM, standardized for Portuguese population between 6 to 11 years old is one of the most used instruments of assessment instruments of non-verbal intelligence. Consisting of 36 items, in which the subject is required to indicate the correct target among six alternatives. Grouped into three sets (A, Ab, B) of 12 items in each one, the CPM solving requires different abilities, namely, A series, requiring visual perceptual ability, AB Series, symmetric ability and B Series, conceptual and analogic thought abilities. It was applied to each schoolchildren (separated in small groups, never exceeding eight participants for session). Each session took approximately ’20 to ’45 min.
The WISC-III (Portuguese version) is an individually administered test of intellectual ability for children aged 6 to 16 years. It consists of 10 mandatory and 3 optional subtests that combine to yield Verbal, Performance, and Full Scale IQ. The WISC-III is an individual test that does not require reading or writing. Verbal subtests are oral questions without time limits, except for Arithmetic. Performance subtests are nonverbal problems, all of which are timed and some of which allow bonus points for extra fast work. The subtests that compose WISC-III, are: 1. Information; 2. Similarities; 3. Arithmetic; 4. Vocabulary; 5. Comprehension; 6. Digit Span; 7. Picture Completion; 8. Coding A and Coding B; 9. Picture Arrangement; 10. Block Design; 11. Object Assembly; 12. Symbol Search and, 13. Mazes. Each session took approximately ’60 to ’120 min.
All the 256 participants were considered on the initial analysis (using CPM). Still, since the distribution of CPM total scores for the study population was similar (Kruskal-Wallis test, p=0.117) across the categories of iodine intake status [determined according WHO's ranks (2007b)], we applied the complete form of WISC-III (with 13 subtests) on the 2 subgroups with 30 children each, selected from the study population as follows: a group of children with Adequate Iodine Nutrition (100-199 ug/L) and a group of children with Moderate Iodine Deficiency (20-49 ug/L). In this study there was a total of 82 children with Moderate Iodine Deficiency and 51 with Adequate Iodine Nutrition. We have selected 30 children in each sub-groups as it represents 37% of the children with Moderate Iodine Nutrition and 58% Adequate Iodine Nutrition. These groups were matched for gender, age and other general socioeconomic characteristics.
Considering this, the authors decided to express only the results related with the two created sub-groups. If the reviewer requires more detailed information on the general study population, this can be provided.
- In the sub-set, especially the moderate iodine deficiency, are there any data demonstrating any gender differences in performance? Any data presented will be warranted.
Differences in brain physiology between sexes do not necessarily relate to differences in intellect. Nonetheless, recent studies conclude that men on average have higher intelligence than women by 3-5 IQ points. In this study, where gender was evenly represented (48.8% of the children were boys), no significant differences were found between the distribution of UIC ranks across gender (χ2(4)=5.285, p=0.209) .
- Fig 2, why moderate iodine deficiency (20-49 ug/L) was chosen and not the mild iodine deficiency (50-99 ug/L) group while your Median UIC was 66.2 ug/L? You have already 43.4% (n=111) of student who falls into the mild iodine deficiency category (Fig 2), in essence, this is the group that needs more attention to show any impairment due to insufficient iodine nutrition.
While the median UIC was 66.2 μg/L, corresponding to a mild iodine deficiency (UIC 50-99 µg/L), the interquartile range was 42.51 (37.38 - 79.89 as 25th - 75th percentile). Also, for comparison, is better to use groups that have clearly separated iodine status, to assure that there are no overlaps due to the proximity of some UIC borderline values in the mild deficient group to the one with adequate iodine intake. If we had chosen randomly 30 children from the mild iodine deficiency, we could have included children with borderline UIC. Also, the Moderate iodine deficiency group represents 32% of the studied children and this deficiency is far more difficult to correct than in children with mild deficiency, therefore this groups requires more attention.
- Would it be appropriate to assign a UIC value to each CPM IQ level in Fig 3 or data overlapping each other as the data set is small, n=30?
The CPM was applied in small groups but to each schoolchildren, thus we can relate the CPM outputs with their UIC. Also, CPM IQ total and percentile scores had no significant differences between iodine status groups (p=0.117 and p=0.384, respectively) as presented in Supplementary Material 2. On table 3, it is displayed that the study subgroups showed no significant differences in CPM IQ total (t(58)=1.961, p=0.055) and percentile scores (t(58)=1.758, p=0.089) between moderate iodine deficiency and adequate iodine intake; however, a clear tendency for schoolchildren to score lower with moderate iodine deficiency can be observed.
- Tables 2 and 3 should have gender data included if it does not cause too much extra analysis. Parents, educators, and policymakers are all interested to ask this question.
In this study, no significant differences were found between the distribution of UIC ranks across gender (χ2(4)=5.285, p=0.209), this is why we decided not the include data regarding gender (see also the answer given in 2.)

Reviewer 2 Report
Excellently designed and performed study with a few minor flaws such as relatively small number (30) of children in each subgroup and repeating the results in the discussion.
Author Response
The authors thank the reviewer comments.
Reviewer 3 Report
In this manuscript, the authors investigated the relationship between Iodine deficiency and cognitive function assessment among elementary school children. By analyzing the urinary iodine concentration and the cognitive performance among children. Authors built up the connections between iodine intake and cognitive performance. Even though the iodine deficiency is not dominant during education performance, this research is still informative for the audience. I recommend to accept this manuscript.
Author Response

(The authors gave the same response as above.)
